METHODS AND RESOURCES

# A single-cell spatiotemporal transcriptomic atlas of mouse prefrontal cortex maps dynamics of intratelencephalic neurons during postnatal development

Hu Zheng[1,2☉], Keji Yan[1,2☉], Xiaojuan Gou[1,2], Zhongchao Wang[1,2], Liyao Yang[1,2], Yayu Huang[3], Huazhen Liu[1,2], Jinxia Dai[1,2*], Leqiang Sun[1,2*], Gang Cao[3*]

**1** State Key Laboratory of Agricultural Microbiology, Huazhong Agricultural University, Wuhan, China, **2** College of Veterinary Medicine, Huazhong Agricultural University, Wuhan, China, **3** Faculty of Life and Health Sciences, Shenzhen University of Advanced Technology, Chinese Academy of Sciences, Shenzhen, China

☉ These authors contributed equally to this work.
* jxdai@mail.hzau.edu.cn (JD); sunlq@mail.hzau.edu.cn (LS); caog@siat.ac.cn (GC)

## Abstract

In early postnatal brain, the prefrontal cortex (PFC) remains immature and highly plastic, particularly for the intratelencephalic (IT) neurons. However, the spatiotemporal molecular and cellular dynamics of PFC during this period remain poorly characterized. Here, we performed spatiotemporal single-cell RNA analysis on mouse PFC during different postnatal time points and systematically delineated the molecular and cellular dynamics of mouse PFC during early postnatal development, among which IT neurons exhibit most dramatic alterations. Based on these comprehensive spatiotemporal atlases of PFC, we deciphered the time-specific molecular and cellular characteristics during the maturation process of IT neurons in PFC, particularly the dynamic expression programs of genes regulating axon development and synaptic formation, and the risk genes of neurological developmental diseases. Furthermore, we revealed the dynamic neuron-glia interaction patterns and the underlying signaling pathways during early postnatal period. Our study provided a comprehensive resource and important insights for PFC development and PFC-associated neurological diseases.

## Introduction

The prefrontal cortex (PFC) serves as the central regulatory hub for higher cognitive functions in the mammalian brain [1,2]. Its sophisticated neural networks critically depend on the precisely coordinated spatiotemporal development of neurons [3,4]. Unlike other cortical regions, the PFC exhibits a prolonged maturation period that extends well into postnatal life [5,6]. While neuronal fate is predominantly determined

**Data availability statement:** The main data supporting the results in this study are available within the paper and its Supporting information files. The raw scRNAseq data are available from GEO (GSE298260). The raw stereo-seq data are available from https://doi.org/10.12412/BSDC.1699433096.20001. The ISH data are available from https://mouse.brain-map.org/. The processed data ready for exploration can be accessed and downloaded via our interactive browsers at https://huggingface.co/spaces/TigerZheng/PFCdev-web (S7 Fig). All data were analyzed with standard programs and packages. The codes were freely accessible via Zenodo at https://doi.org/10.5281/zenodo.17964214.

**Funding:** This work was supported by the National Natural Science Foundation of China (https://www.nsfc.gov.cn/) (32221005 to G.C., 32171022 to JX.D., 31900746 to LQ.S., 32401246 to G.C.), the China Postdoctoral Science Foundation (https://www.chinapostdoctor.org.cn/) (2019M660182 to LQ.S.), the Special Funds Project for Strategic Emerging Industries of Shenzhen Municipal Development and Reform Commission (https://fgw.sz.gov.cn/) (XMHT20240215002 to G.C.), the Key Areas Special Projects for Ordinary Higher Education Institutions in Guangdong Province (https://edu.gd.gov.cn/) (2024ZDZX2010 to G.C.), and the Yunnan Provincial Department of Science and Technology Science and Technology Program Project (https://kjt.yn.gov.cn/) (202503AP140014 to G.C.). The funders had no role in study design, data collection and analysis, decision to publish, or preparation of the manuscript.

**Competing interests:** The authors have declared that no competing interests exist.

**Abbreviations:** ADHD, attention deficit hyperactivity disorder; ANO, anorexia nervosa; ASD, autism spectrum disorder; BP, bipolar disorder; CT, corticothalamic; DEGs, differentially expressed genes; GAM, Generalized Additive Model; GO, gene ontology; IT, intratelencephalic; MDD, major depressive disorder; NPCs, neural progenitor cells; OCD, obsessive-compulsive disorder; OPCs, oligodendrocyte precursor cells; PCA, principal component analysis; PFA, paraformaldehyde; PFC, prefrontal cortex; PT, pyramidal tract; scRNA-seq, single-cell RNA sequencing; SCZ, schizophrenia; TFs, transcription factors; TS, tourette syndrome; UMAP, Uniform Manifold Approximation Projection; WTA, Whole Transcriptome

by prenatal gene expression patterns, the postnatal developmental processes—including ongoing neuronal migration, axonal elongation, and synapse formation—are fundamentally essential for establishing mature neural circuits [7–9]. Notably, early sensory experiences, environmental changes, and stress exposure during this period can significantly influence PFC circuit development and functionality [10–12]. For example, maternal separation during the first postnatal week reduces the number of inhibitory neurons and synapses in mouse PFC, resulting in social deficits in adulthood [13].

Like other cortical regions, PFC follows the conserved inside-out neurogenesis pattern during embryonic development, where neurons are generated in successive waves from deep to superficial layers [14,15]. Corticothalamic (CT) and pyramidal tract (PT) neurons, which reside predominantly in deep cortical layers, are born early and almost complete their radial migration prenatally. These neurons are among the first to establish long-range connections with thalamic and subcortical targets [16–18]. In contrast, intratelencephalic (IT) neurons, the most abundant excitatory neuronal population in the cerebral cortex, distributed across both deep and superficial cortical layers, with their migration and maturation extending into the postnatal period [14,19,20].

The first two postnatal weeks constitute a crucial period for IT axon growth and circuit formation: axons typically reach their target regions within the first week and undergo subsequent refinement through activity-dependent pruning [9,21]. Consequently, this period is essential for the proper establishment of IT-mediated dynamic neural circuits. Despite recent advances in characterizing the molecular diversity of adult PFC IT neurons [22,23], the transcriptional dynamics and regulatory mechanisms governing their axonal growth and synaptic integration during early postnatal period (1–2 weeks) remain poorly understood. Elucidating these mechanisms is crucial for understanding how early molecular programs shape PFC functional architecture and how their disruption may contribute to neurodevelopmental disorders such as autism and schizophrenia.

To delineate the molecular and cellular trajectories underlying postnatal development of PFC neurons and investigate the mechanisms governing early neural circuit assembly, we performed high-throughput single-cell RNA sequencing (scRNA-seq) on mice PFC across four timepoints (postnatal day: P1, P4, P10, and adulthood: P84). Our comprehensive transcriptomic atlas captured dynamic gene expression patterns during postnatal development of neuronal maturation. By integrating these data with spatial transcriptomic profiles from juvenile and adult PFC, we systematically mapped the spatiotemporal distribution patterns of diverse neuronal subtypes. We analyzed in detail the stage-specific molecular characteristics during the maturation process of IT neurons in different layers of PFC and the dynamic expression programs of genes that regulate axon development and synaptic formation. We further revealed the developmental patterns of neuron-glia interaction network and the enrichment patterns of risk genes for neurological diseases across PFC subtypes. This comprehensive dataset provided a valuable resource for studying the development of mouse PFC, and can be viewed online through our user-friendly website: https://huggingface.co/spaces/TigerZheng/PFCdev-web.

## Results

### Single-cell spatiotemporal transcriptomic atlas of developing and mature mouse PFC

To investigate the cell types and gene expression patterns during mouse PFC development, we performed scRNA-seq on mouse PFC at postnatal days P1, P4, P10, and adult (P84) (Fig 1A). These time points were selected because previous studies have shown that the first two postnatal weeks in mice involve multiple developmental events, including axon extension, dendritic branching, and circuit assembly [9,21]. We analyzed the scRNA-seq data using Seurat [24], after quality control and doublet removal, a total of 54,763 (10,522 for P1; 17,169 for P4; 14,063 for P10; and 13,009 for Adult) high-quality cells (average 4,389 genes per cell) were retained for downstream analysis.

To identify the shared and unique cell types in PFC across four postnatal time points, we performed unsupervised clustering on these cells after removing batch effects using Harmony [25]. A total of 21 PFC cell subtypes were identified (Figs 1B and S1A), each defined by top differentially expressed genes (DEGs) and previously reported cell type marker genes (S1B Fig and S1 Table) [22,23]. The distribution of these cell subtypes on the Uniform Manifold Approximation Projection (UMAP) plot suggested potential developmental trajectories (Fig 1C). Notably, we observed abundant immature IT neurons (highly expressed neurodevelopmental genes such as *Cd24a*) in P1 and P4 mice PFC [26], whereas mature IT neurons (highly expressed vesicular glutamate transporter genes such as *Slc17a7*) were predominantly enriched in P10 and adult mice PFC [27,28] (Figs 1D, 1E, and S1C). All IT neurons in the PFC of postnatal mice expressed layer-specific marker genes (*Cux2*, *Rorb*, *Etv1*, and *Npy*), suggesting that although they had already migrated to specific layers, they had not yet fully matured at early postnatal days (P1 and P4). Allen ISH results of developing mouse brain further confirmed the existence of these immature IT neurons (Figs 1D, S1C, and S2D).

Moreover, we observed the proportions of IT neuronal subtypes changed significantly during postnatal development, whereas other neuron subtypes such as L5 PT, L6 CT, L5 NP, and inhibitory neurons (Lamp5, Pvalb, Sst and Vip) were already present and remained relatively steady at P1 stage (Figs 1F, S1B, and S1E), reflecting the sequential birth and development of deep-layer neurons followed by superficial-layer neurons, which is consistent with previous reports on other cortical development [12,15]. Non-neuronal cells (Astro, Microglia, Oligo, OPC) were rare at birth, primarily existing as neural progenitor cells (NPCs). Then their amounts were gradually increased with age, which is consistent with the period of gliogenesis after birth [14].

To further analyze the cell subtypes with their spatial organization, we collected previously published spatial-omics data from sagittal sections of mice at P1 and P77 [29], and extracted the data of PFC region for spatial transcriptomic analysis (S1F Fig). We mapped the cell subtypes from our scRNA-seq atlas onto the spatial coordinates of the P1 (Figs 1G, 1H, and S1G) and P77 (Figs 1I, 1J, and S1H) stereo-seq datasets using Cell2location [30]. The spatial mapping results confirmed that immature IT neurons were predominantly observed in P1, which already exhibited layer-specific localization, while in adult, there were mainly mature IT neurons (Fig 1H and 1J). Other neuronal subtypes, including L5 PT, L6 CT, L5 NP, and inhibitory neurons have been already appeared at P1 stage. This spatial localization further supports the dynamic developmental progression of PFC neurons from early postnatal immaturity to adult maturity.

### Transcriptomic dynamics of mouse PFC neurons during postnatal development

During the first two postnatal weeks of mice, neurons in PFC undergo multiple developmental events, including axon elongation, dendritic branching, and circuit assembly [9]. These events are regulated by the expression of specific transcriptomic programs. Thus, we further analyzed the transcriptomic dynamics of 34,050 excitatory neurons and 7,025 inhibitory neurons from our scRNA-seq data (Fig 2A). To investigate the correspondence of neuronal subtypes at different ages, we applied a supervised classification framework to identify temporal associations among neuronal subtypes based on transcriptomic similarity. Our analysis revealed distinct transcriptomic dynamics among different neuronal subtypes (Figs 2B and S2A–S2C). IT neurons displayed the highest dynamic score across ages, suggesting significant transcriptional

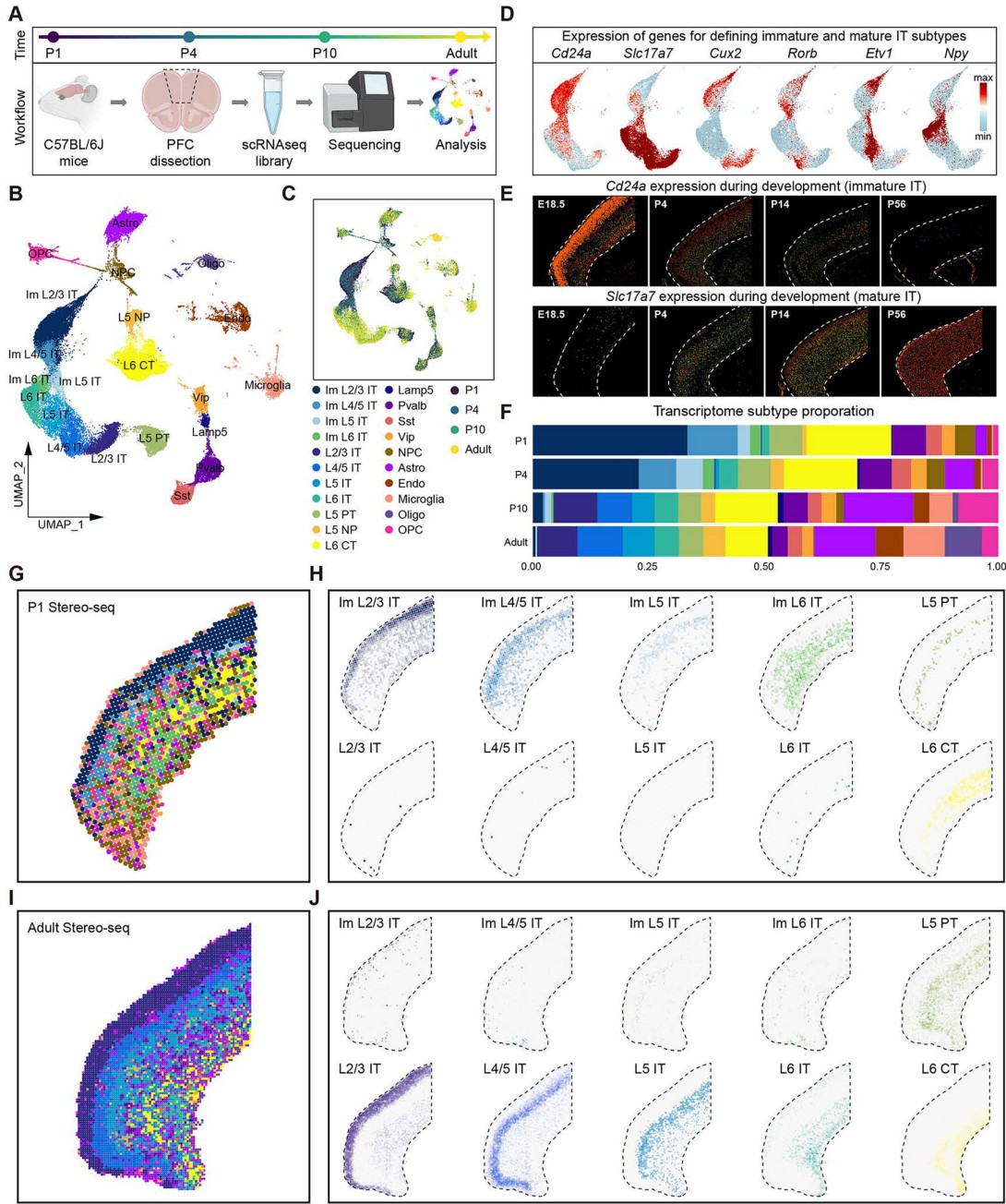

**Fig 1. Spatiotemporal molecular and cellular atlas of mouse PFC during postnatal development. (A)** Schematic diagram of experimental workflow, created with BioRender.com. C57BL/6J mice at four time points (P1, P4, P10, and Adult) were selected, and the PFC region of the brain was dissociated. Then, scRNAseq libraries construction and sequencing were performed, followed by downstream analysis. **(B)** Uniform Manifold Approximation Projection (UMAP) of all cells in mouse PFC from scRNA-seq data, which is colored by cell subtypes. Im, immature. **(C)** UMAP of all cells in mouse PFC from scRNA-seq data, which is colored by time stages. **(D)** Expression of genes for defining immature and mature IT subtypes in UMAP. **(E)** Allen mouse brain ISH images of *Cd24a* (immature IT, top) and *Slc17a7* (mature IT, bottom) genes at E18.5, P4, P14, and P56. **(F)** Proportion of each cell subtype at each developmental stage. **(G)** Spatial distribution of all cell subtypes in mouse PFC from P1 stereo-seq data. **(H)** Spatial distribution of each excitatory neuronal subtype in mouse PFC from P1 stereo-seq data. **(I)** Spatial distribution of all cell subtypes in mouse PFC from adult stereo-seq data. **(J)** Spatial distribution of each excitatory neuronal subtype in mouse PFC from adult stereo-seq data.

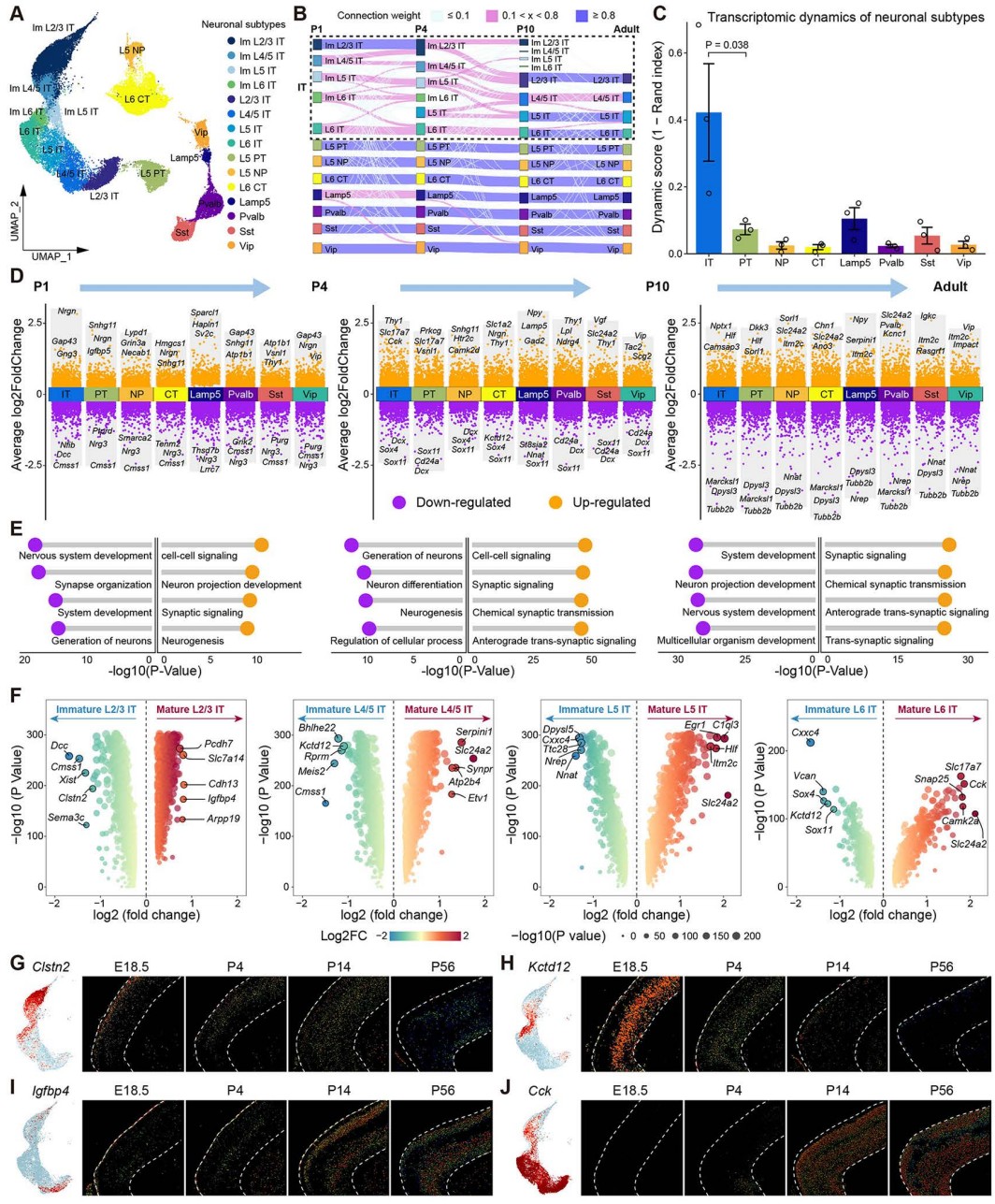

**Fig 2. Transcriptomic dynamics of mouse PFC neurons during postnatal development. (A)** Uniform Manifold Approximation Projection (UMAP) of all neurons in mouse PFC from scRNA-seq data, which is colored by neuronal subtypes. **(B)** Sankey diagram shows temporal associations among mouse PFC neuronal subtypes across ages identified by transcriptomic similarity. Nodes, individual PFC neuronal subtypes at each age; edges, colored based on transcriptomic similarity. **(C)** Transcriptomic dynamics of each neuronal subtype between each pair of consecutive ages, quantified based on transcriptomic similarity. Dynamic score (1 − Rand index (RI)) ranges from 0 (low variation) to 1 (high variation). Bar heights, mean score computed across pairs of consecutive ages; error bars, standard deviation; P = 0.038 for IT against PT. IT neurons are merged into one subtype. The data underlying this Figure can be found in S1 Data. **(D)** Volcano plots show the differentially expressed genes (DEGs) of each neuronal subtype between different ages. The neuronal subtypes of each age are compared with the same neuronal subtypes of the previous age. Orange, up-regulated genes; purple, down-regulated genes. **(E)** Top 4 gene ontology (GO) terms for DEGs in each age comparison. Orange, GO terms of up-regulated genes; purple, GO terms of down-regulated genes. **(F)** Volcano plots show the DEGs between immature and mature IT neurons, separated by different layers. Color represents log2(fold change), point size represents -log10(P value). **(G)** UMAP visualization of *Clstn2* gene expression in IT neurons of mouse PFC (left). Allen mouse brain ISH images of *Clstn2* gene at E18.5, P4, P14 and P56 (right). **(H)** UMAP visualization of *Kctd12* gene expression in IT neurons

of mouse PFC (left). Allen mouse brain ISH images of *Kctd12* gene at E18.5, P4, P14 and P56 (right). **(I)** UMAP visualization of *Igfbp4* gene expression in IT neurons of mouse PFC (left). Allen mouse brain ISH images of *Igfbp4* gene at E18.5, P4, P14 and P56 (right). **(J)** UMAP visualization of *Cck* gene expression in IT neurons of mouse PFC (left). Allen mouse brain ISH images of *Cck* gene at E18.5, P4, P14 and P56 (right).

reprograming in PFC IT neurons during postnatal development (Figs 2C and S2C). In contrast, PT, NP, CT, and inhibitory neurons displayed significantly low dynamic score across ages, indicating that their transcriptomic subtypes remained largely consistent (Figs 2C and S2B).

Next, we investigated how genome-wide expression profiles of individual neuronal subtypes change during development. To this end, we identified DEGs of each neuronal subtypes between different time points based on scRNA-seq data (Figs 2D and S2D). Gene ontology (GO) enrichment analysis revealed that genes related to neuronal development and synaptic organization were abundant in the first two postnatal weeks, and genes related to synaptic signaling transmission were up-regulated concomitant with development (Fig 2E).

Given that IT neurons undergo a developmental transition from immature to mature subtypes, we further analyzed the DEGs of layer-specific IT neurons during postnatal development (Fig 2F and S2 Table). Immature IT neurons predominantly express genes associated with neuronal development. For example, *Clstn2*, which plays crucial roles in nervous system development and synaptic plasticity [31,32], is highly expressed in immature L2/3 IT neurons at P1, but significantly reduced concomitant with development (Fig 2G and S2E). Moreover, *Kctd12*, a key gene affecting neuronal excitability and synaptic transmission by regulating voltage-gated potassium channels [33], is highly expressed in immature L6 IT neurons at P1 and P4, but hardly expressed in adult (Figs 2H and S2F). In contrast, mature IT neurons predominantly express genes associated with neuronal vesicular transport. For example, *Igfbp4* is highly expressed in the mature L2/3 IT neurons at P14 (Figs 2I and S2G). *Cck* gene encodes cholecystokinin protein belonging to the neuropeptide family, is highly expressed in all mature IT neurons in P14 and adult mice (Figs 2J and S2H).

## Identification of cell-type-specific transcription factors during IT neuron postnatal development

Our scRNA-seq data revealed significant transcriptomic dynamics of PFC IT neurons during postnatal development. To further investigate the heterogeneity of IT neurons during development, we reconstructed the developmental trajectory of PFC IT neurons using Monocle2 [34] (Fig 3A and 3B). Pseudotime analysis delineated the dynamic progression from immature IT neurons to mature IT neurons (Fig 3C). IT neurons at different developmental timepoints were orderly distributed along the pseudotime axis. IT neurons at P1 and P4 predominantly located at the beginning of the trajectory, while those at P10 and adult predominantly located at the end of the trajectory (S3A–S3D Fig), suggesting that IT neuron maturation is tightly synchronized with temporal progression. We observed some L6 IT neurons appear early in pseudotime. This is because deep neurons develop first, so some L6 IT neurons have already matured at P1. Through differential gene expression analysis, we identified 3,198 genes that exhibited significant alterations along pseudotime (S3 Table). These genes were clustered into five distinct modules. GO enrichment analysis revealed the function of each gene module (Fig 3D). Genes highly expressed at the beginning of the trajectory were associated with neuronal development and projection organization, whereas those enriched at the end of the trajectory were associated with cell communication and transport.

The development of neurons and their projection organization are under the control of transcription factors (TFs) [4]. Using SCENIC [35], we identified distinct regulons in each IT neuronal subtype (S3E Fig and S4 Table). Based on these results, we delineated a spatiotemporal TFs landscape during IT neuron maturation (Fig 3E). We found that IT neurons were regulated by distinct TFs during different mature and immature states. Immature IT neurons at P1 were primarily regulated by neurodevelopmental TFs, such as the *Sox* gene family, whereas mature IT neurons at adult were mainly regulated by metabolic and signaling transmission-related TFs, such as Nr1d1(+), Hes1(+), Rxrg(+). Of note, these TFs

**Fig 3. The maturation dynamics and the underlying transcription factor regulation of mouse PFC IT neurons during postnatal development.**
**(A)** Uniform Manifold Approximation Projection (UMAP) of mouse PFC IT neurons at different ages during development, which is colored by pseudo-time value. **(B)** Pseudotime trajectory of IT neurons, which is colored by time stages. **(C)** Pseudotime trajectory of each IT neuron subtype, which is colored by IT subtypes. **(D)** Heatmap shows gene modules associated with pseudotime and gene ontology terms for each gene modules. **(E)** Schematic illustrates the spatially specific TFs regulation of IT neurons in the immature (left) and mature (right) mouse prefrontal cortex (PFC). **(F)** Dotplot shows the specific TF regulons expressed in each IT neuron subtype. **(G)** The TF regulatory networks of relguons Sox11(+), Sox12(+), Atf4(+), Tbr1(+) and their top 20 target genes are shown respectively. Edges are colored based on importance score. The TF binding motifs are shown on the left. **(H)** UMAP visualization of *Sox11* gene expression in IT neurons (left). Allen mouse brain ISH images of *Sox11* gene at E18.5, P4, P14, and P56 (right). **(I)** UMAP visualization of *Sox12* gene expression in IT neurons (left). Allen mouse brain ISH images of *Sox12* gene at E18.5, P4, P14, and P56 (right). **(J)** UMAP visualization of *Atf4* gene expression in IT neurons (left). Allen mouse brain ISH images of *Atf4* gene at E18.5, P4, P14, and P56 (right). **(K)** UMAP visualization of *Tbr1* gene expression in IT neurons (left). Allen mouse brain ISH images of *Tbr1* gene at E18.5, P4, P14, and P56 (right).

exhibit layer-specific spatial expression patterns, underlying the specific regulation of the development IT neuronal sub-types within precise locations at different developmental stages (Fig 3E and 3F).

We further analyzed four TFs (Sox11(+), Sox12(+), Atf4(+), and Tbr1(+)) that regulate immature IT neurons in different cortical layers, and constructed their downstream target gene regulatory networks. These TFs bind to target genes through specific binding motifs to exert regulatory functions (Fig 3G). The spatial expression patterns of these TFs were further validated by Allen ISH data. *Sox11*, which has been previously reported in the regulation of circuit development of L2/3 IT neurons in the motor cortex [15], is highly expressed in immature L2/3 IT neurons at P1 (Fig 3H). Similarly, immature L4/5 IT neurons highly express *Sox12* (Fig 3I), whereas *Atf4* and *Tbr1* are expressed in immature L5 IT and immature L6 IT neurons, respectively (Fig 3J and 3K). These findings provided new insights for understanding the transcriptional logic of IT neuron postnatal development.

## Dynamics of circuit wiring molecules during IT neuron postnatal development

Cadherins and axon guidance molecules, play crucial roles in neural circuit wirings [36]. Using scRNA-seq, we analyzed the differential expression of 371 cadherins and axon guidance genes during postnatal development, and identified numerous molecules show developmental stage-specific expression patterns (Fig 4A and S5 Table). Furthermore, we compared the differentially expressed cadherins and axon guidance genes in different IT neuronal subtypes during development (Figs 4B and S4A). For example, *Dcc*, a cell membrane receptor that plays important roles in axon guidance of neurons [37], is highly expressed in immature L2/3 IT neurons at P1. In contrast, the cadherin *Camk2a*, which influences learning and memory processes by regulating neurotransmitter release and synaptic plasticity [38], is highly expressed in mature L2/3 IT neurons at Adult (Figs 4B and S4A).

These circuit wiring molecules are often co-expressed in neuronal subtypes, achieving co-regulation of neurons in the form of gene co-expression modules [39]. To explore the association of these molecules with neuronal development, we constructed a gene co-expression network of cadherin and axon guidance genes using hdWGCNA [40], and identified five distinct co-expression modules (Figs 4C and S4B). We found that these gene modules exhibit distinct neuronal subtype enrichment and development-specific expression patterns. For example, Module 1 (M1) was mainly expressed in immature L2/3 IT neurons, while Module 2 (M2) was expressed in all immature IT neurons. Both M1 and M2 were highly expressed at P1, gradually reduced by adulthood. GO enrichment analysis revealed that these modules were primarily associated with axonogenesis and nervous system development functions (Fig 4D). Module 3 (M3) was highly expressed in immature IT neurons at P4 and P10 and was associated with synaptic development functions. Module 4 (M4) was mainly expressed in mature IT neurons of adult mice. Module 5 (M5) was primarily highly expressed in L6 CT neurons and remained relatively stable during postnatal development (Fig 4D).

We subsequently conducted further analysis of the hub genes in each gene module. M1 highly expressed genes including *Cntn2*, *Clstn2*, *Robo2*, *Plxna4*, and *Unc5d* (Fig 4E and 4F). The cell adhesion molecule *Cntn2* played a crucial role in axonal connectivity and nervous system development [41]. Allen ISH results confirmed its expression in immature L2/3 IT neurons during early postnatal stages (Fig 4G). *Lrp8*, *Igsf3*, *Sema4g*, *Ncam1*, and *Cdh2* were highly expressed in M2 (S4C and S4D Fig). The axon guidance molecule *Sema4g*, preferentially expressed in all immature IT neurons during early postnatal development (S4E Fig), was associated with early neuronal axon formation [42]. M3 expresses high-level of *Efnb3*, *Cdh13*, *Sdc3*, *Cd200*, and *L1cam* (S4C and S4D Fig). Allen ISH results demonstrated that *Cdh13* showed low expression during early postnatal and adult stages, but high expression at P4 and P14 (S4F Fig). M4 highly expressed *Camk2a*, *Clstn3*, *Clstn1*, *Nptn*, and *App* (S4C and S4D Fig). Allen ISH results validated that *Camk2a* exhibited low expression at birth but high expression in adulthood (S4G Fig). M5 specifically expressed *Lrrtm2*, *Cdh11*, *Rgma*, *Ephb1*, and *Islr2* (S4C and S4D Fig). These results suggested that different cadherins and axon guidance molecules may contribute to the construction of neural circuits at different postnatal developmental stages with specific spatial patterns.

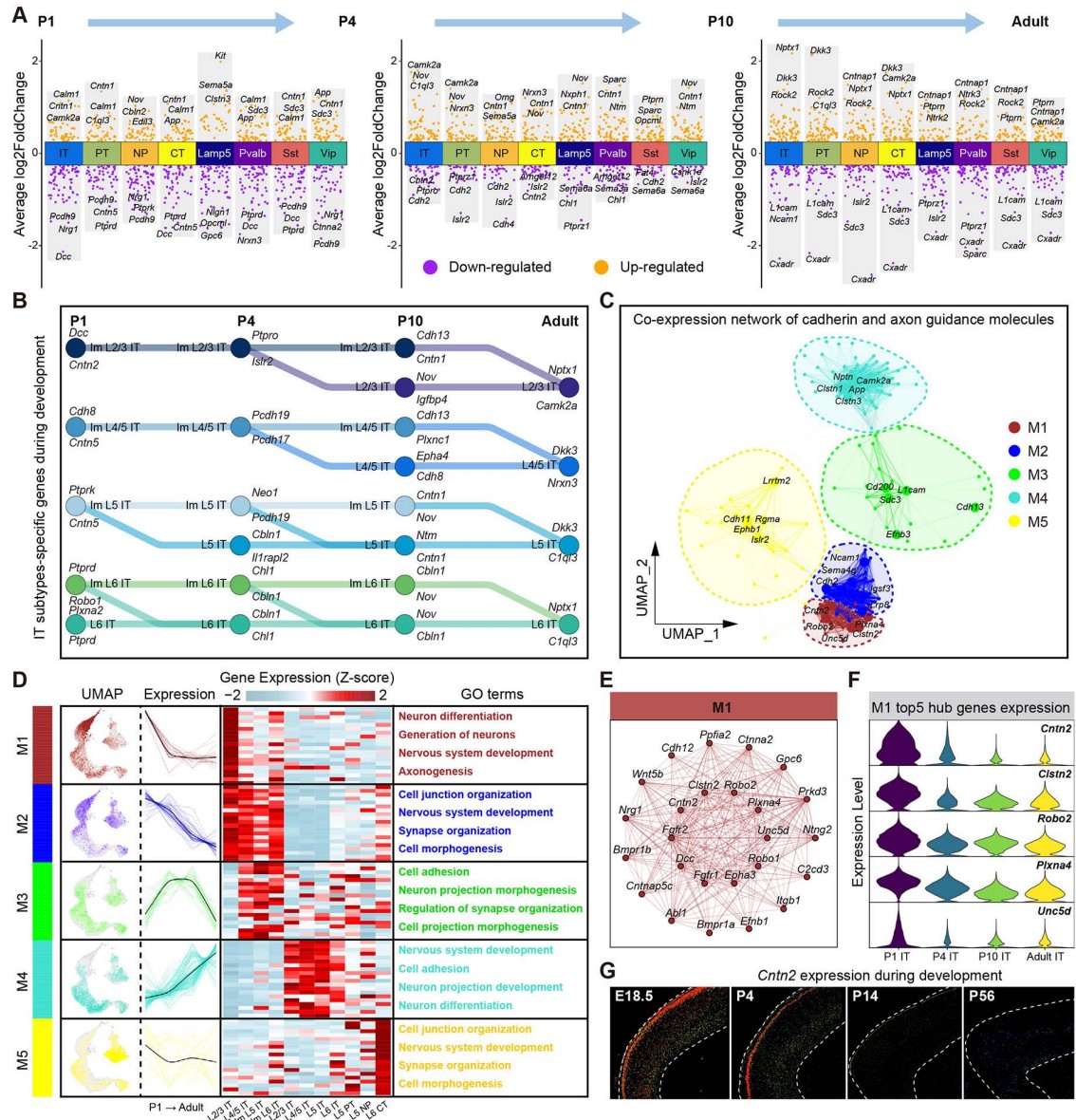

**Fig 4. Expression dynamics of cadherin and axon guidance genes in mouse PFC during postnatal development. (A)** Volcano plots show the differentially expressed cadherin and axon guidance genes of each PFC neuronal subtype between different ages. The neuronal subtypes of each age are compared with the same neuronal subtypes of the previous age. Orange, up-regulated genes; purple, down-regulated genes. **(B)** IT neuron subtypes-specific cadherin and axon guidance genes during development. Nodes, individual IT neuron subtypes at each age; edges, cell developmental trajectories, colored by IT subtypes. Top 2 cadherin and axon guidance genes specifically expressed in each node are displayed. **(C)** Co-expression network of cadherin and axon guidance genes. Each node represents a single gene, and edges represent co-expression links between genes. Genes are divided into five co-expression modules. The top 5 hub genes per module are labeled. **(D)** The expression distribution of each gene module on Uniform Manifold Approximation Projection (UMAP) (left), the expression change curve with age (middle left), the expression heatmap of top 20 hub genes in each neuronal subtype (middle right), and the gene ontology (GO) terms for each module are shown, respectively. **(E)** Co-expression network diagram for module 1. **(F)** Violin plot shows the expression level of module 1 top 5 hub genes in all IT neurons at different ages. **(G)** Allen mouse brain ISH images of *Cntn2* gene at E18.5, P4, P14, and P56.

PLOS Biology

## Interactions between glial cells and IT neurons during postnatal development

Glial cells play active roles in neuronal development, such as synapse formation and myelination [43,44]. To investigate the potential influences of glial cells on PFC neuron development during postnatal development, we firstly reconstructed gliogenesis trajectories using Monocle2 (Fig 5A and 5B). Pseudotime analysis further revealed distinct developmental trajectories among different glial subtypes (Fig 5C). The early postnatal stages (P1 and P4) contained abundant NPCs and oligodendrocyte precursor cells (OPCs), located at the beginning of the pseudotime trajectory. These cells subsequently differentiate into astrocytes and oligodendrocytes [45]. Some NPCs positioned later in the pseudotime axis is because NPCs still exist in the P10 age, reflecting the plasticity of neurons in the first two weeks after birth [7,8]. Mature astrocytes and oligodendrocytes were located at the end of the trajectory and exhibited different developmental trajectories. Glial cells from different ages showed orderly distribution along the pseudotime axis, indicating that gliogenesis is tightly synchronized with temporal progression (S5A–S5D Fig).

Through differential gene expression analysis, we identified 2,441 genes that exhibited significant changes along pseudotime (S6 Table). These genes were clustered into six distinct modules. GO enrichment analysis revealed the function of each gene module (Fig 5D). Genes highly expressed at the beginning of the trajectory were associated with metabolic process and nervous system development, whereas those enriched at the end of the trajectory were associated with cell transport and myelination. These findings suggest that the development of glial cells might be tightly synchronized with the maturation of neurons.

To further investigate the cell–cell interactions among glial subtypes and IT subtypes, we employed CellChat [46] to infer intercellular communication networks at four postnatal developmental stages (Fig 5E). The analysis revealed strong communication of astrocytes with IT neurons at P1, which gradually reduced by adulthood. Oligodendrocytes showed higher communication with IT neurons at P4 and P10, but much weaker in adulthood. OPCs maintained consistently strong communication with IT neurons across all developmental stages, while microglia demonstrated relatively weak communication at every stage. Subsequently, we explored the ligand-receptor pairs that showed significant age-dependent variations (Figs 5F and S5E–S5H). At P1, astrocytes expressed the ligand gene *Slit2*, which may interact with receptor genes *Robo1* and *Robo2* in IT neurons, activating the SLIT signaling pathway (S5E and S5I Fig). *Slit-Robo* signaling is well known for axon repulsion during nervous system development [47]. At P4 and P10, oligodendrocytes expressed the ligand gene *Sema3d*, which may engage with receptor genes including *Nrp1* and *Plxna1* in IT neurons, activating the SEMA3 signaling pathway (Figs 5H, 5I, and S5G). Ligand-receptor spatial colocalization analysis further validated that *Sema3d-Plxna1* genes exhibited strong interactions at P10, but significantly reduced at Adult (Figs 5J, 5K, and S5J).

Collectively, these findings demonstrated that glial cells may play crucial regulatory roles during the development of IT neurons. The temporally specific communication patterns and distinct signaling pathways employed by different glial subtypes suggest their possible specialized functions in guiding neuronal maturation and circuit formation during postnatal development.

## Enrichment of neurological developmental disease risk genes in mouse PFC during postnatal development

PFC is known to be associated with numerous neurological diseases [48]. Deciphering the cell subtype-specific and developmental stage-dependent expression patterns of neuropsychiatric risk genes is fundamental to elucidating disease pathogenesis. To this end, we investigated the enrichment of risk genes for eight neurological disorders in diverse PFC cell subtypes during postnatal development using scDRS [49] and GWAS summary statistics from previous studies (S7 Table) [50–58]. These diseases include attention deficit hyperactivity disorder (ADHD), anorexia nervosa (ANO), autism spectrum disorder (ASD), bipolar disorder (BP), major depressive disorder (MDD), obsessive-compulsive disorder (OCD), schizophrenia (SCZ), and tourette syndrome (TS). Overall, risk genes associated with different diseases tend to be enriched in specific cell subtypes of mouse PFC (Fig 6A). For example, risk genes linked to ADHD, ASD, BP, SCZ, and TS are primarily enriched in excitatory neuronal subtypes, while risk genes associated with OCD are mainly enriched in

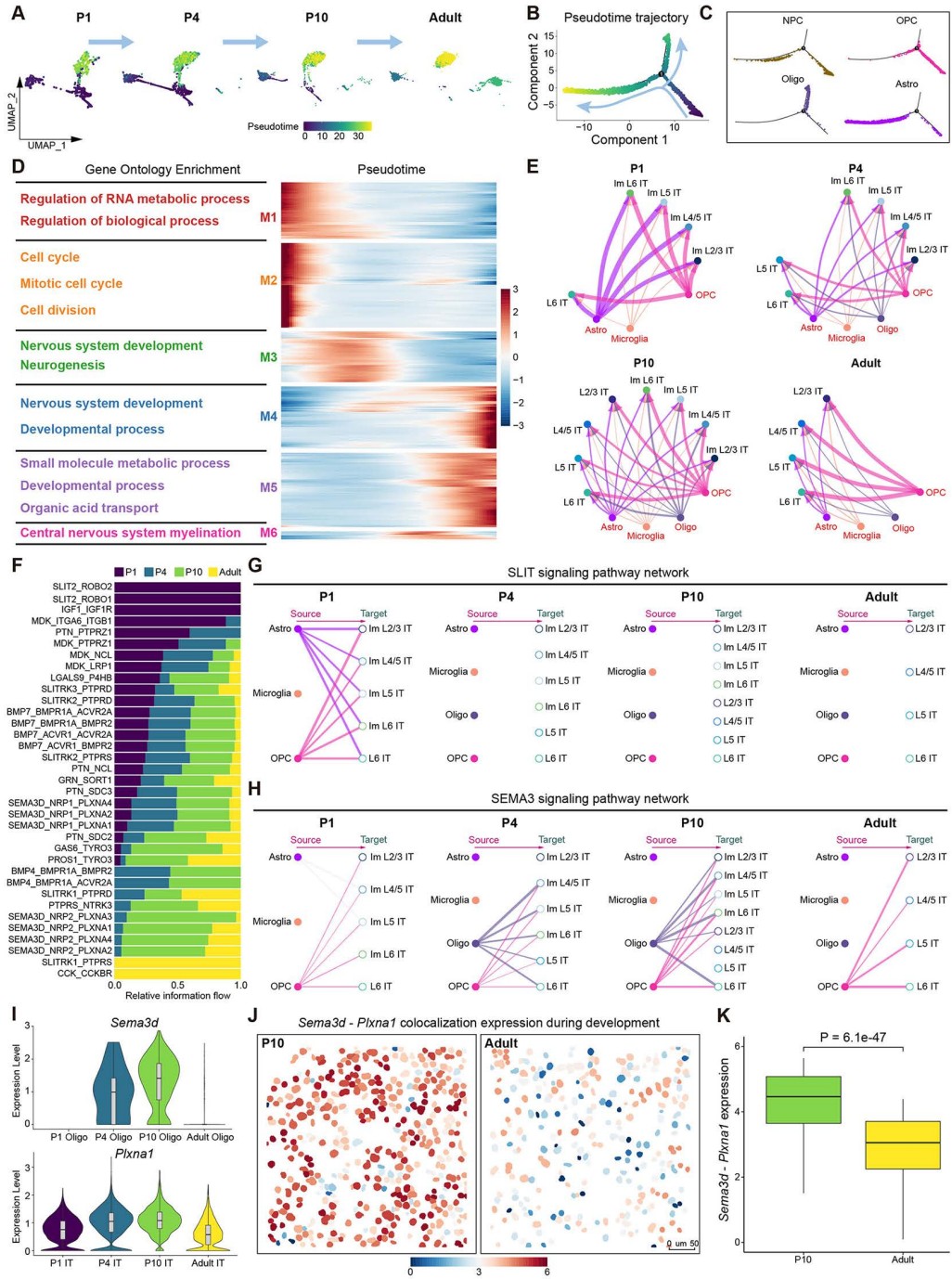

**Fig 5. Gliogenesis dynamics and their communications with IT neurons in mouse PFC during postnatal development. (A)** Uniform Manifold Approximation Projection (UMAP) of glial subtypes at different ages during postnatal development, which is colored by pseudotime value. Microglia are excluded due to different developmental origins. **(B)** Pseudotime trajectory of glial cells, which is colored by pseudotime value. **(C)** Pseudotime trajectory of each glial subtype, which is colored by glial subtypes. **(D)** Heatmap shows gene modules associated with pseudotime and gene ontology terms for each gene modules. **(E)** Cell–cell communication network among glial subtypes and IT neuron subtypes at different postnatal stages. **(F)** Signaling pathways are ranked based on differences in overall information flow across ages in the inferred networks. **(G)** The inferred SLIT signaling pathway network among glial subtypes and IT neuron subtypes at different ages. **(H)** The inferred SEMA3 signaling pathway network among glial subtypes and IT subtypes at different postnatal ages. **(I)** Violin plots show the expression levels of the ligand gene *Sema3d* in oligodendrocytes (top) and the receptor

gene *Plxna1* in IT neurons (bottom) at different postnatal stages. **(J)** Visualization of co-localization expression of *Sema3d* and *Plxna1* in P10 (left) and Adult (right) mice PFC after cell segmentation. Color indicates the intensity of co-localization expression. **(K)** Quantitative comparison of *Sema3d* and *Plxna1* co-localization expression in P10 and Adult mice PFC. The data underlying this Figure can be found in S1 Data.

inhibitory neuronal subtypes. And risk genes associated with MDD are enriched in both excitatory and inhibitory neuronal subtypes. ANO-related risk genes are mainly enriched in microglia subtype (Figs 6A, 6C, and S6H). These findings emphasize the cell type specificity of neurological disease risk, suggesting that genetic susceptibility may converge on certain cell subtypes during postnatal stages.

We further analyzed the temporal expression patterns of disease-associated risk genes in each cell subtype enriched for different neurological diseases. The top 1,000 GWAS disease risk genes by weight score were used for analysis (S8 Table). Differential gene expression analysis suggested that these disease risk genes may function at different developmental ages and specific cell subtypes (Figs 6B and S6A–S6G). For example, MDD risk genes were primarily enriched in neuronal subtypes, which agreed with previous study indicating that alterations in gene expression of glutamatergic and GABAergic neurons may lead to MDD symptoms [59]. ANO risk genes were enriched in microglia subtype, showing distinct enrichment patterns (Fig 6C and 6D). Previous studies have reported a reduction in the number of microglia in ANO mice [60]. *Tcf4* and *Egr1* are two risk genes for MDD, and their expression is reduced in MDD patients according to previous studies [61,62]. They show different temporal expression patterns. Our data indicated that *Tcf4* in IT neurons was mainly expressed at P1, while *Egr1* was predominantly expressed at adult (Fig 6D and 6E). Allen ISH results further validated the expression patterns of these genes across different postnatal ages (Fig 6F). Together, these results depict how risk genes for various neurological diseases are enriched in specific cell subtypes at different postnatal time points, helping to understand the causes of diseases and discover therapeutic targets.

## Discussion

In this study, we systematically dissected the dynamic changes in cell types and transcriptional profile during postnatal development of the mouse PFC by integrating scRNA-seq data and spatial transcriptomic data from different developmental time points. Our data demonstrated that while IT neurons in early postnatal mice (P1 and P4) have already migrated to specific cortical layers, they are not fully mature. Through spatiotemporal expression analysis of marker genes and spatial mapping of cell subtypes, we found that immature IT neurons already exhibit layer-specific distribution at P1, but their transcriptomic profiles were significantly different from those of mature IT neurons in adulthood. This result supported the view that "neurons require additional transcriptomic and functional maturation after completing migration" [15]. Notably, other neuronal subtypes (e.g., L5 PT, L6 CT, and inhibitory neurons) show relatively consistent transcriptomic profiles during postnatal development, consistent with the view that deep-layer neurons mature first and superficial-layer neurons mature later during cortical development [12].

Our data also provided an in-depth investigation of dynamic expression patterns of TFs and circuit wiring molecules in PFC IT neurons during postnatal development. We found that these TFs have spatiotemporal expression patterns, which bind to target genes through specific binding motifs to form complex regulatory networks. By constructing gene co-expression networks, we discovered that axon guidance molecules and cell adhesion molecules exhibit modular expression patterns across different developmental stages. These TFs regulate molecules such as axon guidance cues and play crucial role in neuronal development and circuit formation [3,4]. These precise temporally regulated expression patterns reflect dynamic molecular requirements during neural circuit assembly and refinement [41,42], while also providing important clues for understanding the pathogenesis of neurodevelopmental disorders [38,63].

Glial cells (e.g., astrocytes, microglia, and oligodendrocytes) play crucial roles in the development of neurons through dynamic cell–cell communications during postnatal PFC development. These spatiotemporal interactions are mediated by specific signaling molecules secreted by specific glial cells at different stages, which elegantly regulate the developmental

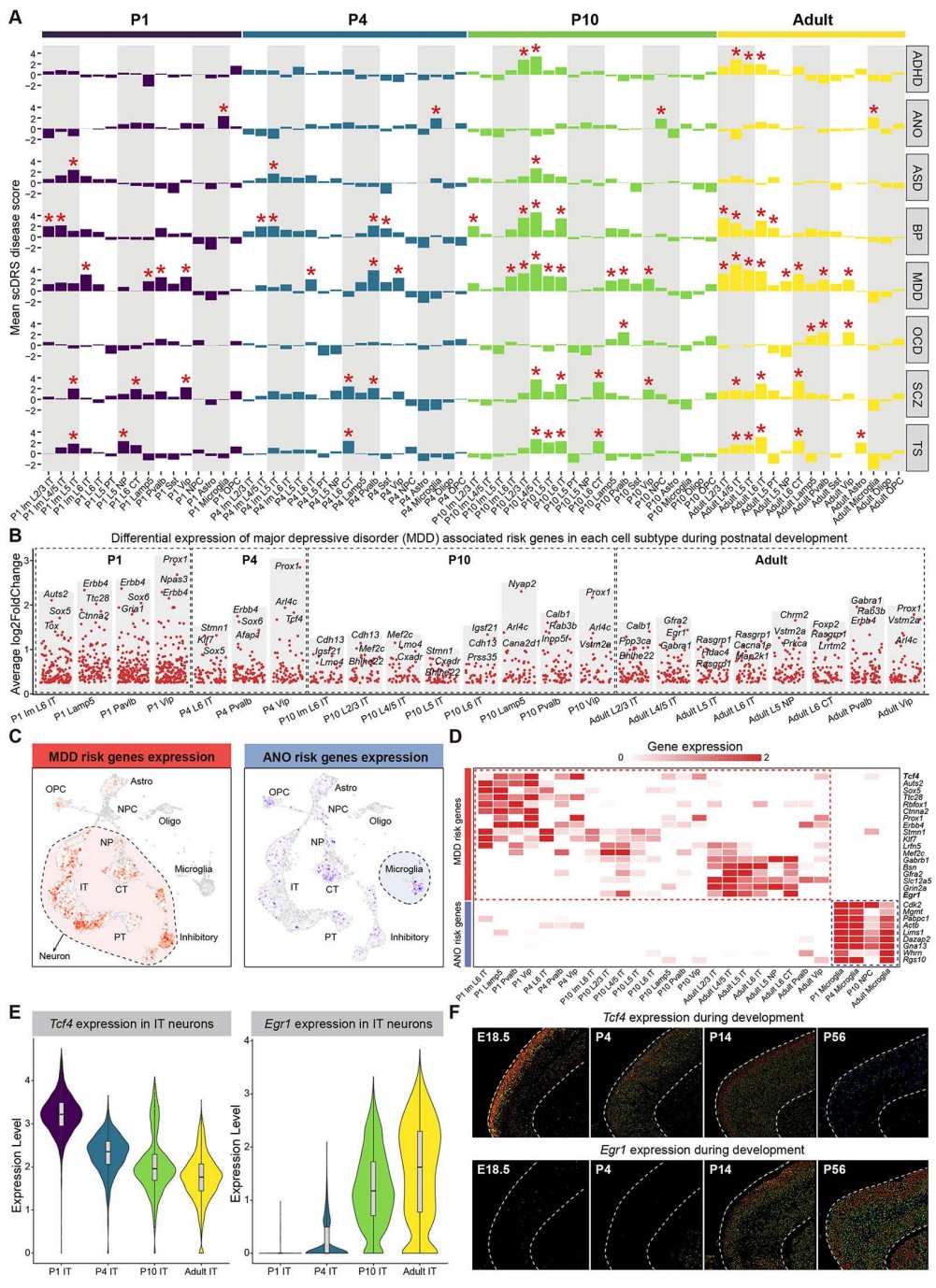

**Fig 6. Enrichment analysis of neurological developmental disease risk genes in diverse cell subtype of mouse PFC during postnatal development. (A)** Barplot shows the enrichment levels of GWAS candidate genes from 8 PFC-related disease in different cell subtypes. The bar heights represent mean disease score calculated by scDRS, annotated with time stages and cell subtypes. Asterisks denote the cell subtypes with $P$-value <0.05. ADHD, attention deficit hyperactivity disorder; ANO, anorexia nervosa; ASD, autism spectrum disorder; BP, bipolar disorder; MDD, major depressive disorder; OCD, obsessive-compulsive disorder; SCZ, schizophrenia; TS, tourette syndrome. **(B)** Volcano plot shows the differential expression of MDD risk genes across different postnatal stages in subtypes marked with asterisks in **(A)**. The top 3 risk genes are displayed in each subtype. **(C)** Uniform Manifold Approximation Projection (UMAP) visualization of the MDD (left) and ANO (right) scDRS disease score. **(D)** Heatmap shows the expression of MDD and ANO risk genes in their enriched subtypes. **(E)** Violin plot shows the expression level of *Tcf4* (left) and *Egr1* (right) genes in MDD-enriched IT neurons at different ages. **(F)** Allen mouse brain ISH images of *Tcf4* (up) and *Egr1* (bottom) genes at E18.5, P4, P14, and P56.

processes of neurons such as synapse formation and myelination [43,44,64]. Our intercellular communication analysis revealed several important signaling pathways (such as SLIT and SEMA3) may mediate glia-IT neurons interaction, which would play important roles in axon guidance and fasciculation of PFC IT neurons [47,65]. These findings revealed the active and dynamic regulatory roles of glial cells during IT neuron maturation. The precise spatiotemporal interactions between glial cells and neurons likely serve as critical safeguards for proper neural network assembly.

The postnatal PFC is vulnerable to neurodevelopmental and neurological diseases, many of which are associated with specific cell subtypes and developmental stages [48]. For example, IT/PT imbalance is a cause of many neurological diseases such as autism, schizophrenia, and depression [66]. Abnormalities in CT neural circuits may lead to diseases such as schizophrenia and bipolar disorder [67]. Disruption of the development or function of GABAergic interneurons may lead to epilepsy and other neuropsychiatric disorders associated with cognitive impairment [68]. In this study, we systematically analyzed the enrichment patterns of GWAS risk genes for 8 PFC-associated neurological diseases across different cell subtypes during postnatal development [50–58]. Our data revealed that disease risk genes exhibit distinct cell-type-specific and developmental-stage-specific expression profiles. For example, MDD risk genes tend to be enriched in neuronal subtypes, while ANO risk genes tend to be enriched in Microglia subtype, which is consistent with previous study [59,60]. And MDD risk genes show different temporal expression patterns. This discovery provided critical insights for understanding the pathogenesis of neurological diseases, suggesting that genetic susceptibility may contribute to disease onset by affecting specific cell subtypes during distinct developmental stages. Future research could further investigate the functional mechanisms of these risk genes within identified cellular subtypes and elucidate how they collectively lead to neural circuit dysfunction, thereby advancing the translation from genetic associations to mechanistic understanding.

In conclusion, our data revealed the spatiotemporal molecular and cellular dynamics of the mouse PFC during postnatal development. However, further research with more age stages single-cell and spatial transcriptome data is needed for a more comprehensive understanding of PFC maturation. And the functions of some key TFs, circuitry molecules, and signaling pathways also require further experimental investigation. Additionally, all analyses in this study were performed on mice. Although some studies have reported certain conservation of neocortical cell types across mammalian species [29,69,70], considering the timing differences in PFC development among different species, especially between mice and human, it remains unclear whether the cell types and gene expression dynamics during PFC development in mice are also applicable to other species.

## Methods

### Ethical statement

All animal procedures in this study were strictly conducted in accordance with the Guideline for Ethical Review of Animal Welfare (GB/T 35892-2018) of the People's Republic of China. All animal experiments (HZAUMO-2025-0270) were conducted according to protocols approved by the Scientific Ethics Committee of Huazhong Agricultural University, Hubei, China. The brain tissue for single-cell RNAseq was obtained from mice at different postnatal ages: P1, P4, P10, P84. Animals were housed in standardized cages with a 12 h:12 h light:dark cycle with unrestricted access to food and water.

### Single-cell dissociation

All mice used for scRNA-seq were female. Two mice at each time point were used for single-cell sequencing. Mice were anesthetized on ice then its brain tissue was immediately sectioned into 250 μm slices in ice-cold ACSF (124 mM NaCl, 2.5 mM KCl, 1.2 mM $NaH_2PO_4$, 24 mM $NaHCO_3$, 5 mM HEPES, 13 mM glucose, 2 mM $MgSO_4$, and 2 mM $CaCl_2$, pH: 7.3–7.4) on vibratome (Leica VT1200). Slices containing PFC region were transferred into Petri dish containing ice-cold ACSF with 45 μM Actinomycin D (Sigma-Aldrich, Cat# A1410). PFC tissue was isolated under a microscope then quickly cut into small pieces less than 1 mm and transferred to digestion buffer containing 3 mg protease XXIII (Sigma-Aldrich, P5380) and 30 U/ml papain (Sigma-Aldrich, P3125). The digestion was performed at 34 °C for 20 min and bubbled with a mixture gas

of 95% $O_2$ and 5% $CO_2$ continuously. After the digestion, the tissue was transferred to stop buffer (ACSF contain 1 mg/ml Trypsin Inhibitor (Sigma-Aldrich, T6522), 2 mg/ml BSA (Sigma-Aldrich, A2153), and 1 mg/ml Ovomucoid Protease Inhibitor (Worthington, LK003153). The digested tissue was titrated with 4 polished Pasteur pipets, of which the bore diameter of the pipets is successively decreasing from 600 to 150 μm. Following trituration, suspension was filtered through a 30 mm filter, then centrifuged at 300g for 5 min. The pellet was then resuspended in ice-cold, carbogen-bubbled ACSF with 0.01% BSA to reach a final concentration of 40–50 cells per microliter and then subject to scRNA-seq library preparation using BD Rhapsody single-cell Analysis System (BD Biosciences, 633702) according to the manufacturer's manual.

## Single-cell RNA sequencing library preparation

Single-cell capture and library preparation were performed by the BD Rhapsody Single-Cell Analysis System (BD Biosciences, USA). Briefly, single-cell suspension was loaded into a BD Rhapsody cartridge (BD Biosciences, 633733), and single-cell mRNA capture was achieved by the cartridge with >200,000 microwells and magnetic beads (BD Biosciences, 664887) with barcoded capture oligos. Then, beads were collected for subsequent cDNA synthesis and library construction following the BD Rhapsody single cell 3′ whole transcriptome amplification (WTA) workflow. Finally, the libraries were quantified using the Agilent 2100 Bioanalyzer (Agilent, USA) and the Qubit 4.0 (Thermo Fisher Scientific, USA) and were sequenced on Illumina NovaSeq 6000 (Illumina, USA) with 300-bp reads (150-bp paired-end reads).

## RNA FISH tissue section preparation

Mice were perfused transcardially with 4% paraformaldehyde (PFA). The harvested brains were post-fixed in 4% PFA for 24 h at 4 ℃ and then cryoprotected by immersion in 30% sucrose until they sank. Subsequently, the brains were embedded in OCT compound and sectioned into 15-μm-thick coronal sections using a Leica cryostat (CM3050 S). Sections containing the PFC were mounted onto poly-L-lysine-coated glass slides and stored at −80 ℃ until further use. For the hybridization procedure, the sections were fixed in 4% PFA for 10 min, permeabilized with pre-cooled methanol at −80 ℃ for 15 min, and digested with pepsin (2 mg/mL; Sigma-Aldrich, P0525000) at 37 ℃ for 90 s.

## RNA FISH in situ hybridization

The probe design and hybridization procedures were performed according to the Mip-seq protocol [71]. The detailed steps are as follows: Probe Design and Preparation: Briefly, 22 pairs of padlock probes were designed for each target gene based on its mRNA length (all probe sequences are listed in S9 Table). Each padlock probe contains 13-nucleotide (nt) sequences at both ends that are complementary to the target mRNA. The middle region of the padlock probe comprises two repeats of the sequence complementary to the fluorescent detection probe. An initiator primer was designed with its 5′ end complementary to the 3′ end of the target sequence and its 3′ end complementary to the padlock probe. Padlock probes were phosphorylated using T4 Polynucleotide Kinase (200 μM; Vazyme, N102-01) and subsequently annealed with initiator primer.

Hybridization Procedure: (I) Pretreatment: brain sections were fixed (4% PFA, 10 min), permeabilized (pre-cooled methanol, −80 ℃, 15 min), and digested with pepsin (2 mg/mL, 37 ℃, 90 s). (II) Hybridization: after washing with PBSTR and 4× SSC, sections were hybridized with the probe mixture overnight at 37 ℃. (III) Ligation: sections were incubated with a ligation mixture containing SplintR ligase (1 U/μL) at 25 ℃ for 2 h. (IV) RCA: following a wash, sections were incubated with an RCA mixture containing Phi29 polymerase (1 U/μL) at 30 ℃ for 6 h. (V) Detection and Imaging: RCA products were detected with fluorescent probes (37 ℃, 30 min), counterstained with DAPI, and imaged on a Leica THUNDER Imager (20× objective).

## scRNA-seq data pre-processing

Raw reads were pre-processed using the BD Rhapsody Whole Transcriptome Amplification (WTA) analysis pipeline (v1.11) (https://bd-rhapsody-bioinfo-docs.genomics.bd.com). The R1 reads were analyzed to identify the cell label sequences (CLS), common linker sequences (L), and Unique Molecular Identifier (UMI) sequence. The R2 reads were

used for aligning to the reference genome and annotating genes. For WTA reference genome, we selected GRCm38-PhiX-gencodevM19-20181206.tar file. For transcriptome annotation, we selected gencodevM19-20181206.gtf file. After setting up, we ran pipeline using the default parameters. The expression matrix file generated by pipeline was used for downstream transcriptome analysis.

### scRNA-seq quality control

Single-cell RNA-seq transcriptome analysis is mainly performed by R package Seurat (v4.4.0) [24]. Briefly, Seurat object was created using the "CreateSeuratObject" function, and the gene expression profile of each cell was then normalized using the "NormalizeData" function with scale.factor = 10000. We filtered the following cells: nCount_RNA < 1000, nFeature_RNA < 1000, and mitochondrial contents > 15%. The following genes were filtered: min.cells < 3, mitochondrial genes, and ribosomal genes. Then, the R package DoubletFinder (v2.0.3) [72] was used to remove potential doublets.

### Clustering of scRNA-seq transcriptome

Firstly, Harmony (v1.2.0) [25] was used to remove batch effects between four samples. The "RunHarmony" function was applied to integrate the four datasets. After integration, we performed the standard Seurat clustering analysis workflow. We used the "ScaleData" function to scale the integrated data, and performed principal component analysis (PCA) using the "RunPCA" function. Then we computed the nearest neighbors used the "FindNeighbors" function with top 30 PCs. The "FindClusters" function was used for clustering analysis with resolution = 0.5. Then, the clusters were annotated based on previously reported markers of PFC cell types [23]. We manually removed some mixed low-quality cell clusters that expressed markers of multiple cell types. Twenty-one cell subtypes were annotated. Then, we ran the UMAP dimensional reduction using the "RunUMAP" functions, and visualized data using functions provided by Seurat. In total, our data contains a transcriptome expression matrix of 54,763 cells and 31,158 genes.

### Stereo-seq spatial spot annotation

We collected previously published stereo-seq data from sagittal sections of mice at P1 and P77 [29], and extracted the PFC region for spatial transcriptomic analysis. To annotate spots in stereo-seq data, we used Cell2location (v0.1.4) [30] to map the annotated information of our scRNAseq subtypes onto the bin50 spots in stereo-seq data. Briefly, we first used "filter_genes" function to filter low-quality genes with parameters: cell_count_cutoff = 15, cell_percentage_cutoff2 = 0.05, nonz_mean_cutoff = 1.12. The signatures were estimated from scRNAseq data to account for potential batch effect using "setup_anndata" function with default parameters. Then we used Cell2location function to create and train a cell2location spatial mapping model with parameters: N_cells_per_location = 1, detection_alpha = 200. Finally, we exported the estimated cell subtypes abundance, and take a cell subtype with the highest abundance score as the identity for each spot in the stereo-seq data. The Endo subtype was excluded from subsequent analyses, due to its abnormal score and was not the focus of this study.

### Supervised classification framework

To identify the associations among neuronal subtypes at different ages, we used XGBoost [73], a gradient boosted decision tree-based supervised classification framework. Three time periods, P1–P4, P4–P10, and P10–Adult were compared. Two thousand highly variable genes were used as input features for XGBoost, and neuronal subtype labels were used as output. We trained XGBoost classifier to learn neuronal subtype labels from the previous age dataset, and used it to classify cells in the next age dataset. The correspondences between the true label of the next age dataset and the label assigned by the XGBoost classifier was used to map neuronal subtypes between different ages. The main steps are as follows: We use the R package xgboost (v1.7.5.1) to implement XGBoost supervised classification. The "xgb.DMatrix"

function was used to construct the xgb.DMatrix object using 2,000 highly variable genes and neuron subtype labels. We used the "xgb.cv" function to perform cross-validation to determine the optimal number of iterations. Then, the XGBoost model was trained using the "xgboost" function. Finally, the "predict" function was used to perform prediction.

## Differential gene expression analysis

To perform differential gene expression analysis, we used the Wilcoxon rank-sum test, a nonparametric test, to compare whether there is a significant difference in the medians of two independent samples. Log2 fold change >0.25 was used as the threshold to determine significant genes. Specifically, we used the "FindAllMarkers" function from the R package Seurat (v4.4.0) to perform differential gene expression analysis with parameters: logfc.threshold = 0.25, test.use = "wilcox".

## Gene ontology enrichment analysis

GO terms enrichment analysis was performed using gProfiler [74] with default parameters. The adjusted *p*-values <0.05 was used as the significance threshold. And the results of GO terms were limited to enrichment in GO biological processes.

## Constructing single cell trajectories

To investigate the development of IT neurons and glia cells, we reconstructed the developmental trajectories using Monocle2 (v2.24.0) [34]. Briefly, CellDateSet object was created using the "newCellDataSet" function, and 2,000 high variable genes were selected to define cellular processes using the "setOrderingFilter" function. Then, "reduceDimension" function was performed to reduce data dimensionality with parameters: max_components = 2, method = "DDRTree". Finally, cells were sorted along the pseudotime trajectories using the "orderCells" function. We visualized the analysis results through the visualization function provided by Monocle2.

## Differential gene expression analysis of genes along pseudotime

The differential gene expression analysis of genes along pseudotime was performed using the Generalized Additive Model (GAM) in Monocle2. Monocle2 assigned each cell a "pseudotime" value, which recorded its progress through the process in the experiment. The GAM can model a gene's expression level as a smooth, nonlinear function of pseudotime. Then, likelihood ratio test was used to identify pseudotime-related genes. The main steps are as follows: We used the "differentialGeneTest" function from the R package Monocle2 (v2.24.0) to find DEGs related to pseudotime with parameter: fullModelFormulaStr = "~sm.ns(Pseudotime)". The *q*-values <0.01 were used as the significance threshold.

Then, we used "ward.D2", a hierarchical clustering algorithm, to cluster pseudotime DEGs into modules. Specifically, we used the "plot_pseudotime_heatmap" function from the R package Monocle2 (v2.24.0) to plot a heatmap of DEGs along pseudotime with parameter: hclust_method = "ward.D2".

## Identifying single-cell transcription factor regulatory network

We used pySCENIC (v0.11.2) [35] package to identify TF regulatory network from scRNAseq data. First, we used the mm_mgi_tfs.txt file to filter out TFs in our data and generated co-expression modules using the GRNBoost2 algorithm. Then, we used "pyscenic ctx" command to prune the initial network based on the relationship between motifs and TFs and the ranking of motifs on gene regulation. Finally, we used "aucell" to identify cells with active gene sets in scRNAseq data.

## Single-cell weighted gene co-expression network analysis

Single-cell weighted gene co-expression network analysis is mainly performed by R package hdWGCNA (v0.3.01) [40]. Briefly, "SetupForWGCNA" function was used to select neural signal molecule or neural circuit wiring molecule genes

for analysis. We used "MetacellsByGroups" function to construct metacell expression matrix, and normalized the matrix using "NormalizeMetacells" function. Then, "SetDatExpr" function was used to specify the expression matrix for network analysis, and soft power threshold was selected using "TestSoftPowers" function. We used "ConstructNetwork" function to construct the co-expression network, and visualized the network using functions provided by hdWGCNA.

### Cell–cell communication analysis

To investigate the interactions among glial subtypes and IT neuron subtypes, we employed CellChat (v2.1.2) [46] to infer intercellular communication networks for each time point using snRNA-seq data. Briefly, "createCellChat" function was used to create a CellChat object, and CellChatDB.mouse databse was selected using "subsetDB" function. Then, we used "computeCommunProb" and "filterCommunication" functions to compute and filter the communication probability between each cell subtypes with parameters: type = "triMean", min.cells = 10. The communication probability of each signaling pathway was computed using "computeCommunProbPathway" function. Finally, "aggregateNet" function was used to calculate the aggregated network.

### Ligand-receptor spatial colocalization analysis

To quantify gene expression, we segmented cells in the DAPI images using cellpose (v3.1.1.1) [75] and then calculated the fluorescence intensity of *Sema3d* and *Plxna1* in each cell as the expression level. Then, we used SpaGene (v0.1.0) [76] to calculate the strongest ligand-receptor interaction between each cell and its five spatial nearest neighbors as the ligand-receptor colocalization expression level. Briefly, "models.Cellpose" function was used to load the cellpose model with parameters: model_type = 'nuclei', gpu = True. Then we used the "model.eval" function to perform cell segmentation with parameters: flow_threshold = 0.4, cellprob_threshold = 0, diameter = 30, min_size = 15. Finally, the "plotLR" function in SpaGene was used to calculate the colocalization expression scores of *Sema3d* and *Plxna1* ligand-receptor.

### Disease risk genes enrichment analysis

The neurological diseases GWAS summary statistics were collected from previous studies (S7 Table) [50–58]. First, we used MAGMA (v1.10) [77] to perform SNP annotation and GWAS disease risk gene weighting. The top 1,000 GWAS disease risk genes by weight score were used for further analysis (S8 Table). Then we used scDRS (v1.0.4) [49] to perform disease risk genes enrichment analysis. Briefly, "scdrs compute-score" command was used to evaluate disease enrichment to individual cells. Then, "scdrs perform-downstream" command was used to obtain the group-level statistics for each cell type. All parameters are set to the default values according to scDRS documentation. The *P*-value < 0.05 was used as the threshold of significance.

## Supporting information

**S1 Fig. Cell subtypes and marker genes of mouse PFC during postnatal development. (A)** Uniform Manifold Approximation Projection (UMAP) visualization of mouse PFC cell subtypes at different postnatal developmental stages. **(B)** Dotplot shows the expression patterns of marker genes in cell subtypes (top). Barplot shows the proportion of different time stages in each cell subtype. **(C)** Violin plot shows the expression patterns of marker genes in IT neuron subtypes. **(D)** Allen mouse brain ISH images of marker genes at E18.5, P4, P14, and P56. **(E)** Bar plot shows the coefficient of variation of the changes in the proportion of different neuronal subtypes during development. **(F)** A full sagittal section of Allen Mouse Brain atlas with the PFC region circled in red color. Allen Mouse Brain Atlas, mouse.brain-map.org and atlas.brain-map.org. **(G)** Spatial distribution of each cell subtype in mouse PFC from P1 stereo-seq data. **(H)** Spatial distribution of each cell subtype in mouse PFC from Adult stereo-seq data.
(TIF)

**S2 Fig. Neuronal subtypes proportion and differentially expressed genes in mouse PFC during postnatal development.** **(A)** Uniform Manifold Approximation Projection (UMAP) of all neurons in mouse PFC from scRNA-seq, which is colored by time stages. **(B)** Line plot shows the proportion of each neuronal subtype during postnatal development. IT neurons are merged into one subtype. The data underlying this Figure can be found in S1 Data. **(C)** Line plot shows the proportion of each IT neuronal subtype during postnatal development. The data underlying this Figure can be found in S1 Data. **(D)** The number of up-regulated and down-regulated DEGs of each neuronal subtype between different postnatal stages. The neuronal subtypes of each stage are compared with the same neuronal subtypes of the previous stage. Orange, up-regulated genes; purple, down-regulated genes. **(E)** Violin plot shows the expression level of *Clstn2* gene in L2/3 IT neurons at different postnatal stages. **(F)** Violin plot shows the expression level of *Kctd12* gene in L6 IT neurons at different postnatal stages. **(G)** Violin plot shows the expression level of *Igfbp4* gene in L2/3 IT neurons at different postnatal stages. **(H)** Violin plot shows the expression level of *Cck* gene in all IT neurons at different postnatal stages.
(TIF)

**S3 Fig. The pseudotime trajectories and transcription factor relguons of mouse PFC IT neuron.** **(A)** Uniform Manifold Approximation Projection (UMAP) of all IT neurons in mouse PFC from scRNA-seq, which is colored by time stages. **(B)** Pseudotime trajectory of IT neurons, which is colored by pseudotime value. **(C)** Pseudotime trajectory of IT neurons, which is colored by pseudotime states. **(D)** Box plot shows the pseudotime distribution in each time stages. The data underlying this Figure can be found in S1 Data. **(E)** Regulon specificity score (RSS) ranking of relguons in each IT neuron subtype.
(TIF)

**S4 Fig. Cadherin and axon guidance genes co-expression modules.** **(A)** Volcano plots show the differentially expressed cadherin and axon guidance genes of each IT neuron subtype at different developmental stages. For each stage, each IT neuron subtype is compared with other IT subtypes. Orange, up-regulated genes; purple, down-regulated genes. **(B)** hdWGCNA dendrogram of the co-expression network of cadherin and axon guidance genes. **(C)** Co-expression network diagram for module 2 to module 5. **(D)** Violin plots show the expression level of top 5 hub genes for module 2 to module 5 at different postnatal stages. **(E–H)** Allen mouse brain ISH images of *Sema4g* (E), *Cdh13* (F), *Camk2a* (G), *Cdh11* (H) gene at E18.5, P4, P14, and P56.
(TIF)

**S5 Fig. Glia cells pseudotime trajectories and their intercellular communication with IT neurons.** **(A)** Uniform Manifold Approximation Projection (UMAP) of all glial cells in mouse PFC from scRNA-seq, which is colored by time stages. Microglia are excluded due to different developmental origins. **(B)** Pseudotime trajectory of glial cells, which is colored by time stages. **(C)** Pseudotime trajectory of glial cells, which is colored by pseudotime states. **(D)** Box plot shows the pseudotime distribution in each time stages. The data underlying this Figure can be found in S1 Data. **(E–H)** Dot plots show significant ligand-receptor interactions among Astrocyte (E), Microglia (F), Oligo (G), OPC (H), and IT subtypes across time points. The dot color indicates the communication probability, and the dot size reflects the $P$ value. The square color indicates time points. **(I)** Violin plots show the expression levels of the ligand gene *Slit2* in astrocytes (top) and the receptor gene *Robo1* in IT neurons (bottom) at different postnatal stages. **(J)** RNA FISH of *Sema3d* and *Plxna1* ligand-receptor genes at P10 (left) and Adult (right) in mice PFC. The small boxes indicate the expression of each gene. Blue: DAPI, green: *Sema3d*, red: *Plxna1*.
(TIF)

**S6 Fig. Expression of neurological disease risk genes in mouse PFC.** **(A–G)** Volcano plots show the differential expression of BP (A), ADHD (B), SCZ (C), TS (D), ANO (E), ASD (F), and OCD (G) risk genes across different postnatal stages in subtypes marked with asterisks in Fig 6A. The top 3 risk genes are displayed in each subtype. **(H)** Uniform Manifold Approximation Projection (UMAP) visualization of the ADHD, BP, ASD, OCD, SCZ, and TS scDRS disease score.
(TIF)

**S7 Fig. Overview of the PFCdev-web.** PFCdev-web is a shisny application that allows users to interactively access our data. Home Page provides information about our project and how to use it interactively. Users can access our scRNAseq data through scRNAseq Page, and access processed stereo-seq data through Spatial Page.
(TIF)

**S1 Table. Differentially expressed genes (DEGs) for each cell subtype.**
(XLSX)

**S2 Table. Differentially expressed genes (DEGs) between immature and mature IT neurons.**
(XLSX)

**S3 Table. Significant genes associated with pseudotime of IT neurons.**
(XLSX)

**S4 Table. Regulon specificity score (RSS) of relguons in each IT subtype.**
(XLSX)

**S5 Table. hdWGCNA co-expression modules of cadherin and axon guidance genes.**
(XLSX)

**S6 Table. Significant genes associated with pseudotime of glia cells.**
(XLSX)

**S7 Table. GWAS summary statistics for 8 PFC-associated neurological diseases.**
(XLSX)

**S8 Table. Top 1,000 GWAS disease risk genes by weight score.**
(XLSX)

**S9 Table. RNA FISH probe sequence.**
(XLSX)

**S1 Data. Additional numerical data for Figs 2C, 5K, S2B, S2C, S3D, and S5D.**
(XLSX)

## Acknowledgments

We thank the core facility of National Key Laboratory of Agricultural Microbiology in Huazhong Agricultural University. We thank Spatial FISH, Co., Ltd. for assistance with single-cell RNA sequencing. The stereo-seq dataset is provided by Brain Science Data Center, Chinese Academy of Sciences (https://braindatacenter.cn/).

## Author contributions

**Conceptualization:** Hu Zheng, Jinxia Dai, Leqiang Sun, Gang Cao.

**Data curation:** Hu Zheng.

**Formal analysis:** Hu Zheng, Xiaojuan Gou.

**Funding acquisition:** Jinxia Dai, Leqiang Sun, Gang Cao.

**Methodology:** Hu Zheng, Keji Yan.

**Project administration:** Hu Zheng.

**Resources:** Hu Zheng, Zhongchao Wang, Liyao Yang, Yayu Huang, Leqiang Sun.

**Supervision:** Huazhen Liu, Jinxia Dai, Gang Cao.

**Validation:** Hu Zheng, Keji Yan, Leqiang Sun.

**Visualization:** Hu Zheng, Xiaojuan Gou.

**Writing – original draft:** Hu Zheng, Keji Yan, Leqiang Sun.

**Writing – review & editing:** Hu Zheng, Keji Yan, Jinxia Dai, Leqiang Sun, Gang Cao.

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
