## [Editor Report · Decision Letter 0]

11 Sep 2025

Dear Dr Zheng,

Thank you for submitting your manuscript entitled "Spatiotemporal molecular and cellular dynamics of intratelencephalic neurons in mouse prefrontal cortex during postnatal development" for consideration as a Research Article by PLOS Biology.

Your manuscript has now been evaluated by the PLOS Biology editorial staff, and I am writing to let you know that we would like to send your submission out for external peer review.

However, I wanted to let you know that we were unable to obtain expert advice from an Academic Editor about the suitability of your paper for PLOS Biology, so we will be looking to the reviewers for enthusiasm.

In addition, we would like to consider your article as a Resource. Please select the "Methods & Resources" article type when you complete the next stage of submission.

Before we can send your manuscript to reviewers, we need you to complete your submission by providing the metadata that is required for full assessment. To this end, please login to Editorial Manager where you will find the paper in the 'Submissions Needing Revisions' folder on your homepage. Please click 'Revise Submission' from the Action Links and complete all additional questions in the submission questionnaire.

Once your full submission is complete, your paper will undergo a series of checks in preparation for peer review. After your manuscript has passed the checks it will be sent out for review. To provide the metadata for your submission, please Login to Editorial Manager (https://www.editorialmanager.com/pbiology) within two working days, i.e. by Sep 13 2025 11:59PM.

Kind regards,

Taylor

Taylor Hart, PhD,

Associate Editor

PLOS Biology

thart@plos.org

---

## [Decision Letter · Decision Letter 1]

12 Nov 2025

Dear Dr Zheng,

Thank you for your patience while your manuscript "Spatiotemporal molecular and cellular dynamics of intratelencephalic neurons in mouse prefrontal cortex during postnatal development" went through peer-review at PLOS Biology. Your manuscript has now been evaluated by the PLOS Biology editors, an Academic Editor with relevant expertise, and by several independent reviewers.

In light of the reviews, which you will find at the end of this email, we are pleased to offer you the opportunity to address the comments from the reviewers in a revision that we anticipate should not take you very long. We will then assess your revised manuscript and your response to the reviewers' comments with our Academic Editor aiming to avoid further rounds of peer-review, although we might need to consult with the reviewers, depending on the nature of the revisions.

As you will see, the reviewers expressed interest in the resource value of the study as well as some of the findings. However, they also raised concerns about several aspects, including methodological concerns, missing information, and limitations in the comprehensiveness of the dataset. After discussing these points with the Academic Editor, we think that you should revise your study to address all of the methodological issues and requested clarifications, and provide a thorough response to all of the reviewers' points.

**IMPORTANT - SUBMITTING YOUR REVISION**

*Resubmission Checklist*

*Published Peer Review*

*PLOS Data Policy*

*Blot and Gel Data Policy*

Sincerely,

Taylor

Taylor Hart, PhD,

Associate Editor

PLOS Biology

thart@plos.org

REVIEWS:

Reviewer #1: In this manuscript, Zheng, Yan, and colleagues present a temporal, single-cell analysis of mouse prefrontal cortex during postnatal development. They show that intratelencephalic neurons exhibit the greatest transcriptomic diversity across postnatal timepoints and characterize molecular and cellular developmental patterns. While the manuscript offers a descriptive and useful resource for mouse prefrontal cortex postnatal development on a single-cell level, it presents few biological insights and most importantly provides insufficient methodological detail.

Major concerns:

- The number and sex of animals used in the experiment is not explicitly stated anywhere. How many mice are included in each age group?

- The manuscript is missing information about several key methods:

1. The supervised classification framework used to identify temporal associations among neuronal subtypes.

2. The method used for differential gene expression analysis and the threshold for significance.

3. The method for GO term analysis and the threshold for significance.

4. Differential gene expression analysis of genes along pseudotime. How was this done? Were the authors comparing differences in gene expression between pseudotime states? This was unclear.

5. Clustering of pseudotime DEGs into modules. How were the genes clustered/how were these modules identified?

- The GWAS risk genes used in this analysis do not fully reflect the current state of knowledge on psychiatric disorders. Given that these are highly polygenic traits with many small-effect variants contributing, it would be more informative to use a method, such as single-cell Disease Relevance Score (scDRS), that uses the latest GWAS summary statistics to provide a more comprehensive view of how polygenic risk maps onto the cell types and developmental stages.

Minor concerns:

- The authors report significant changes in the proportions of IT neurons between developmental stages but have not quantified this. A statistical method that can account for variability between samples (e.g., propeller method) is needed to make this claim.

- It would be helpful to include a full section from the Allen Mouse Brain ISH atlas as reference to indicate where the regions of interest are located.

- The circle volcano plots in Figure 2D and 4A are difficult to interpret without axes. I recommended replacing these figures with plots like those in Fig S2D and Fig S4A.

- It is unclear why mature some L6 IT neurons appear early in pseudotime and some NPCs are positioned later in pseudotime. The authors should clarify this apparent discrepancy.

- Please randomize the plotting order of points in the pseudotime trajectory in Fig S3B so that all ages are clearly visible. Once corrected, this figure would be more informative than Fig 3B and the authors could consider using it as a replacement for Fig 3B.

- It is unclear whether the analysis in Figure 4B is comparing between ages or IT neuron subtypes. If it is the latter, the authors should discuss why several of the same genes (e.g., Cbln1, Chl1, Nov, Cdh13, Cntn1) are shared across several subtypes.

- Are the genes in Module 1 already expressed in deep-layer IT neurons earlier in development (e.g., E18.5)? This would help clarify whether M1 is truly subtype-specific to L2/3 IT neurons or whether it reflects a maturation program that first appears in deeper layers and then becomes enriched in more superficial layers between P1-P4.

- Fig 5K is missing a legend. Please add this to help with interpretation.

Reviewer #2: This study combined scRNA-seq with spatial transcriptomics across multiple developmental stages (P1, P4, P10, P84) to provide a high-resolution spatiotemporal map of neuronal and glial maturation in mouse PFC. The study advances understanding of excitatory neuron subtype development, transcription factor networks, axon guidance programs, and neuron-glia signaling. Addtionally, the study provided enrichment analysis of GWAS risk genes (ASD, SCZ, MDD, etc.) across cell types and developmental windows provides translational relevance, linking developmental PFC molecular programs to human disease risk. However, several concerns reduced the significance of this study.

Major concerns:

1. The study is largely descriptive and with limited mechanistic insight. While transcriptomic and computational analyses are strong, experimental validation (e.g., perturbation of key TFs or signaling pathways) is lacking.

2. Four time points (P1, P4, P10, P84) were used, but the spatial data is missing (P1 and P77 are re-analyzed). This should be clearly stated and with own spatial data generated to fill in the time points. In addition, ideally, intermediate stages (e.g., P14, P21, adolescence) would provide a more continuous trajectory of PFC maturation.

3. CellChat predictions are insightful, but direct evidence (e.g., ligand-receptor validation, co-localization, or functional manipulation) is missing. Additional experimental results are required to support the conclusions.

Minor concerns:

1. Other populations (e.g., PT, CT, interneurons) are noted as stable, but their contributions to PFC development and disease are under-discussed.

2. While code and data are deposited, more explicit instructions or tutorials for reanalysis would enhance utility for the broader community.

3. this study reference human studies, but cross-species relevance (especially timing differences between mouse and human PFC development) should be more explicitly addressed.

---

## [Editor Report · Decision Letter 2]

15 Dec 2025

Dear Dr Zheng,

Thank you for your patience while we considered your revised manuscript "Spatiotemporal molecular and cellular dynamics of intratelencephalic neurons in mouse prefrontal cortex during postnatal development" for publication as a Methods and Resources at PLOS Biology. This revised version of your manuscript has been evaluated by the PLOS Biology editors and the Academic Editor.

Based on our Academic Editor's assessment of your revision, we are likely to accept this manuscript for publication. Please also make sure to address the following data and other policy-related requests.

IMPORTANT: Please ensure that your next revision addresses these editorial points:

-------------------

**Title:

-- We would like to tweak your paper's title to emphasize what we see as its main strengths as a resource. Is the following alternative title acceptable to you?

"A single-cell transcriptomic atlas of mouse prefrontal cortex maps dynamics of intratelencephalic neurons during postnatal development"

**Financial disclosure statement:

-- Please add links to the funding agencies in the Financial Disclosure statement in the manuscript details.

**Ethics:

-- Please include the specific national or international regulations/guidelines to which your animal care and use protocol adhered. Please note that institutional or accreditation organization guidelines (such as AAALAC) do not meet this requirement.

**Data:

-- Thank you for uploading your sequencing data to GEO and for providing DEGs and other related information in the supplement. Please also provide some additional numerical data in a supplemental excel file referred to as S1 Data and with the file name "S1_Data.xlsx". This applies to the data for the following figure panels:

2C

5K

S2BC

S3D

S5D

-- Please also indicate in the figure legends where the relevant underlying data can be found, e.g. “The data underlying this Figure can be found in S1 Data” or “The data underlying this Figure can be found in https://doi.org/10.5281/zenodo.XXXXX”

**Code availability:

-- Thank you for providing the underlying code in GitHub. However, because Github depositions can be readily changed or deleted, please make a permanent DOI’d copy (e.g. in Zenodo) and provide this URL in the manuscript and Data Availability Statement.

-------------------

We expect to receive your revised manuscript within two weeks.

*Published Peer Review History*

*Press*

Sincerely,

Taylor

Taylor Hart, PhD,

Associate Editor

thart@plos.org

PLOS Biology

---

## [Editor Report · Decision Letter 3]

19 Dec 2025

Dear Dr Zheng,

Thank you for the submission of your revised Methods and Resources "A single-cell spatiotemporal transcriptomic atlas of mouse prefrontal cortex maps dynamics of intratelencephalic neurons during postnatal development" for publication in PLOS Biology. On behalf of my colleagues and the Academic Editor, Claude Desplan, I am pleased to say that we can in principle accept your manuscript for publication, provided you address any remaining formatting and reporting issues. These will be detailed in an email you should receive in early January (after the office reopens following the winter holidays) from our colleagues in the journal operations team; no action is required from you until then. Please note that we will not be able to formally accept your manuscript and schedule it for publication until you have completed any requested changes.

PRESS

Sincerely,

Taylor

Taylor Hart, PhD,

Associate Editor

PLOS Biology

thart@plos.org